# Determination of Selenium in Selenium—Enriched Products by Specific Ratiometric Fluorescence

**DOI:** 10.3390/s23229187

**Published:** 2023-11-15

**Authors:** Munire Aimaitiniyazi, Turghun Muhammad, Ayzukram Yasen, Sainawaer Abula, Almire Dolkun, Zulhumar Tursun

**Affiliations:** 1State Key Laboratory of Chemistry and Utilization of Carbon-Based Energy Resources, College of Chemistry, Xinjiang University, Urumqi 830017, China; 107552100876@stu.xju.edu.cn (M.A.); 107552100877@stu.xju.edu.cn (A.D.); 15276554017@163.com (Z.T.); 2Key Lab of Natural Product Chemistry and Application, School of Chemistry and Chemical Engineering, Yili Normal University, Yining 835000, China; 0110212845@ylnu.edu.cn; 3School of Safety Science and Engineering, Xinjiang Engineering Institute, Urumqi 830023, China; sanawar_abla@outlook.com

**Keywords:** ratiometric fluorescence, selenium, determination, internal reference, food safety

## Abstract

Selenium (Se), as one of the essential and nutrient components of living organisms and plants, plays an important role in life activities, while excessive selenium is hazardous to human health. So, the establishment of an effective method for simple, rapid, and highly sensitive determination of selenium content is crucial in the field of food composition analysis and other areas. In this paper, a novel and simple ratiometric fluorescence method for the determination of Se has been developed using 9-anthracenemethanol (AM) as the ratiometric fluorescence reagent on the basis of the conventional fluorometric assay which utilized 2,3-diaminonapthalene (DAN) as fluorescent ligand. The ratiometric method was compared with the conventional method with respect to precision and accuracy. The inter-day and intra-day precisions (RSDs) of the ratiometric fluorescence method ranged from 2.08 to 2.78% and 1.28 to 1.84%, with mean recoveries of 93.2~98.0% and limit of detection (LOD) and limit of quantification (LOQ) of 0.0016 and 0.0049 μg/mL, respectively. This method was successfully applied to the determination of total selenium in selenium-enriched milk and selenium-supplemented shampoo, with the results in agreement with those obtained by inductively coupled plasma mass spectrometer (ICP-MS). The results demonstrated that the precision and accuracy of the ratiometric fluorescence method were superior to those of the conventional fluorescence method, and the interferences of various environmental factors were effectively eliminated. The precision and accuracy of the conventional method can be significantly improved by simply adding an elaborately selected ratiometric fluorescence reagent, and the new method will have broader practical applications.

## 1. Introduction

Selenium (Se) is a vital trace element that plays an essential role in the normal metabolism of plants, animals, and humans [1,2]. The primary source of Se for the human body is dietary intake [3], and for animals, it comes from plants [4]. However, the toxicity, bioavailability, and reactivity of Se depend on its chemical form and concentration [5]. According to the Recommended Daily Intake of Trace Elements for Chinese Residents (WS/T 578), healthy adults should consume 60 μg/day of Se on average, with a maximum daily intake of 400 μg/day [6]. In the United States and Europe, the recommended daily intake levels are 55 μg/day and 70 μg/day, respectively [7]. The Se levels in natural food sources are generally insufficient to meet the body’s physiological demands. Consequently, selenium-fortified foods (e.g., tea and milk) and Se supplements have emerged as the primary modalities for Se supplementation [8]. In recent years, many countries have marketed selenium-enriched foods. However, prolonged consumption of foods containing less than 0.1 mg/kg of Se can result in Se deficiency, whereas those with over 1 mg/kg can cause toxicity [9]. This has raised concerns regarding food safety and quality [10]. Hence, establishing a sensitive, rapid, efficient, and accurate quantitative method for Se detection is crucial to ensure the safety of consumers and maintain the quality standards of selenium-enriched food products. 

Currently, analytical techniques utilized for Se determination in foods include atomic absorption spectrometry [11], atomic fluorescence spectrometry, colorimetric method, electrochemical analysis, chromatography [12], inductively coupled plasma mass spectrometer (ICP-MS) [13], as well as high-performance liquid chromatography (HPLC) and HPLC-hyphenated techniques [14,15]. Although these methods provide high sensitivity and specificity for Se detection, they are expensive, cumbersome, and require trained personnel to operate, making them impractical for routine laboratory use [16,17,18]. In light of the limitations associated with existing analytical techniques for Se detection, there is a pressing need to develop a more straightforward and user-friendly approach. The fluorescence analysis of Se has gained widespread attention due to its notable advantages, such as rapid response, simple handling, excellent selectivity, and high sensitivity. A fluorescent nanoprobe was created with an open structure consisting of a silica nanoparticle core and a coating containing the fluorescent molecule 3,3′-diamino-benzidine (DAB) induced by Se(IV), enabling selective detection of Se(IV) [19]. Feng et al. synthesized an inducible fluorescent ligand derived from rhodamine 6G and used it as a probe for detecting Se(IV), which exhibited remarkable sensitivity and selectivity [20]. Thakur and colleagues have recently proposed a novel approach that involves the application of selenium-induced cross-linking of tin-doped carbon quantum dots (CQDs) for the detection of Se(IV) in aqueous solutions [21]. Similarly, Liao et al. have reported a sensitive and selective fluorescence method based on carbon dots (CDs) for label-free detection of trace selenite in water samples [22]. The preceding approaches offer exceptional selectivity, increased sensitivity, and superior detection limits compared with conventional methods. It is worth noting that the fluorescence signals generated in these methods rely on a single fluorescence emission, which makes them vulnerable to environmental interferences [23]. As a result, false positives or false negatives may occur, leading to overestimation or underestimation of detection outcomes. The present national standard methodology [24] utilizes the reaction of 2,3-diaminonaphthalene (DAN) with Se(IV) and a single emission signal for quantitative fluorescence determination. However, it has been extensively demonstrated that fluorescence sensors with a single emission peak do not exhibit optimal performance [25]. Chen et al. have developed an initial ratiometric fluorescence nanosensor for in situ sensing of SeO_3_^2−^ [23]. This sensor exhibits distinct fluorescence emissions of both Se-DAB and CdTe quantum dots at a single excitation wavelength, enabling dependable and precise ratiometric fluorescence detection of selenium. Therefore, the ratiometric fluorescence method has immense potential in selenium detection. However, the preparation of ratiometric fluorescent probes is laborious, expensive, and time intensive.

In this regard, we proposed a novel, facile, and highly sensitive single reference signal ratiometric fluorescence approach on the basis of a national standard method [24] for the detection of total Se content on selenium-enriched products. The method utilized 2,3-diaminonapthalene (DAN) in an acidic environment to produce 4,5-phenylbenzoselenoxazole (Se-DAN), which exhibits greenish fluorescent properties. To establish a quantitative relationship between fluorescence intensity ratio and Se content, 9-anthracenemethanol (AM) was added as a ratiometric fluorescence reagent. The proposed method utilizes a single excitation wavelength to detect double-emission fluorescence without the need for a specific ratiometric fluorescent probe. The ratiometric fluorescence reagent in solution serves as a reference response signal for establishing the aforementioned quantitative relationship. A schematic representation of both the conventional fluorescence method and the developed ratiometric fluorescence approach is depicted in Figure 1. Our approach represents an integration of a ratiometric fluorescent reagent into the national standard methodology, resulting in enhanced precision, sensitivity, and environmental resilience. Furthermore, the dual-emission fluorescence exhibits a noticeable color change that enables the visual detection of Se.

## 2. Materials and Methods

### 2.1. Materials

Analytical-grade chemicals were procured commercially for the experiment. Cresol Red, Hydroxylamine hydrochloride, and SeO_2_ were purchased from Shanghai Adamas Reagent Co. (Shanghai, China). 9-Anthracenemethanol (AM, 99%) was purchased from J&K Scientific (Beijing, China). 2,3-Diamino-naphthalene (DAN, 99%) was obtained from Tricia Chemical Industry Development Co. (Shanghai, China). Ethylenediaminetetraacetic acid disodium salt (EDTA-2Na, 99%) was obtained from Tianjin Zhiyuan Chemical Reagent Co. (Tianjin, China). DAN was purified with cyclohexane to eliminate any fluorescence, whereas other reagents were used without purification. The 100 mg/mL Se(IV) stock solution was prepared by digestion of SeO_2_ solids and diluted with 1% HCI to the final stock solution of 100 ng/mL, which was utilized throughout the entire study. Milk (Shu Hua, Yili, China) and shampoo (Selsun Gold, Hawthorn East, Australia) samples used in this study were from a local market.

### 2.2. Instrumentation

An electric heating plate (DZTW, Shanghai, China) was utilized to wet digest all the samples. Fluorescence spectral measurements were taken at room temperature using a Fluorescence Spectrophotometer (CRT970, Shanghai, China) featuring a tunable excitation light source. Fluorescence photos were captured using a mobile phone beneath a UV detector (SB-2A, Tianjin, China). An inductively coupled plasma mass spectrometer (ICP-MS, 7900, Agilent, Santa Clara, CA, USA) was used for the qualitative analysis of Se in the milk samples. Microwave digestion was completed using a microwave digestion instrument (TANK eco, Shanghai, China).

### 2.3. Plotting of the Standard Curve

Volumes of 0.0 mL, 0.20 mL, 1.0 mL, 3.0 mL, and 5.0 mL of 100 μg/L Se(IV) solution were transferred into separate 50 mL colorimetric tubes. Then, a (1 + 9) hydrochloric acid solution was added to each tube up to the final volume of 5.0 mL. Subsequently, a 20 mL EDTA mixture solution was added, followed by an adjustment of the pH of each solution to 1.5–2.0 using a (1 + 1) ammonia solution. The resulting solutions showed a light red-orange color. Next, 3.0 mL of DAN reagent (1 g/L) was added, mixed well, and then heated in a boiling water bath for 5 min before being removed and cooled to room temperature. In the following, the resulting mixture was transferred to a separation funnel with cotton stuffed at the bottom, followed by the addition of 4.0 mL of cyclohexane. The contents were vigorously shaken for 2 min and then left to settle for 3 min, with protection from light. After phase separation, the aqueous phase was discarded, and the organic phase was collected with caution to avoid the transfer of any portion of the aqueous phase [24]. Fluorescence spectra were measured under excitation at 376 nm (excitation and emission slit widths of 10 nm). The fluorescence standard curve was plotted between fluorescence intensity at 523 nm and the concentration of selenium. Under the same experimental conditions, 0.15 mL of 100 mg/mL AM was added accurately to 4.0 mL of the cyclohexane extract, and the fluorescence intensities were recorded at 410 nm and 523 nm simultaneously. 

### 2.4. Inter-Day and Intra-Day Precision Experiments 

Three specific concentrations of Se-DAN and Se-DAN + AM (25.0 μg/L, 75.0 μg/L, and 125.0 μg/L) were chosen for assessment. In order to evaluate inter-day and intra-day RSDs, experiments were conducted under the same experimental conditions.

### 2.5. Sample Preparation

The digestion procedure outlined in the National Standard for Food Safety [24] was used for the total selenium content determination. In brief, an appropriate amount of each sample was transferred to a 50 mL digestion tube, and a glass funnel was placed on top of each tube to prevent contamination and evaporation during subsequent heating. The samples were then soaked overnight in a 10 mL digestion mixture consisting of nitric acid and perchloric acid in a ratio of 9:1 (*v*/*v*). Digestion temperature was slowly increased from room temperature to avoid violent boiling and instant compensation for the consumed nitric acid. Precautions were taken to prevent the digestion solution from spilling out of the digestion tube. The heating process was continued until a clear and colorless solution was obtained. When approximately 2.0 mL of solution remained, the digestion tube was removed from the heating plate. After cooling to room temperature, 5.0 mL of hydrochloric acid (6 M) was added to the tube and further heated to reduce Se(VI) to Se(IV) [26] until a clear and colorless digestion solution was obtained. The digestion process was considered complete when white fumes appeared, and approximately 2.0 mL of solution was left. 

### 2.6. Sample Measurement

After digestion, the cooled sample solutions were diluted with 1% HCI to the suitable concentration range and filtered. Blank samples were prepared under identical conditions. Finally, a volume of 2.0 mL digestion solution was transferred into a 50 mL colorimetric tube. Then, 3.0 mL of (1 + 9) hydrochloric acid, 20 mL of EDTA mixture solution, and 3.0 mL of DAN reagent (1 g/L) were added to the tube and mixed thoroughly. Afterward, the mixture was heated in a boiling water bath for 5 min, removed from the water bath, and cooled at room temperature. The mixture was transferred to a separation funnel with cotton stuffed at the bottom and extracted with 4.0 mL of cyclohexane by shaking vigorously for 2 min and kept for 3 min. It needed to be protected from light at this step. The aqueous phase was discarded after phase separation. The organic phase was collected and determined using both the conventional fluorescence method and the proposed ratiometric fluorescence method.

### 2.7. Smartphone Application for the Detection of Se(IV)

Fluorescent color photographs of Se-DAN cyclohexane solutions with different concentrations of Se(IV) were taken by a smartphone (Honor X50, Huawei Technologies Co., Ltd., Shenzhen, China) under 365 nm ultraviolet light (Power: 6 W). Color information (RGB value) was obtained by a smartphone app (color recognizer, Kaifeng Lefan Network Technology Co., Kaifeng, China).

## 3. Results and Discussion

### 3.1. Characterization of the Molecular Structure of Se-DAN

After reacting with DAN, Se(IV) was extracted with cyclohexane and evaporated to obtain the crude product, which was dissolved in petroleum ether by heating and filtered. The filtrate recrystallized at −4 °C in the form of red rod-shaped crystals. The crystals were characterized by HNMR spectroscopy, and the results were in agreement with those reported in the literature [27]. δ H (600 MHz, CDCl_3_) 8.39 (2 H, s), 7.77 (2H, q, J 6.7, 3.2), 7.27 (2H, q, J 6.7, 2.7) (Appendix A).

### 3.2. Standard Curve

The selection of the internal reference fluorescent reagent should fulfill the following conditions: (1) it should be chemically and physically stable, (2) should have a sufficiently strong fluorescence under the excitation wavelength of the analyte, and (3) the emission wavelength should be completely separated from that of the analyte. With the above factors in mind, according to the excitation and emission wavelengths of the analyte (376/520 nm), the commonly used fluorescent reagents, such as fluorescein (505/520 nm), pyrene-butyric acid (345/376 nm), and coumarin (325/450 nm) were excluded. AM was selected because of its good solubility in cyclohexane, the exciting solvent, having an excitation wavelength of 365 nm, which is very close to that of an analyte. In addition, AM has strong fluorescence emission at 410 nm excited with 376 nm, which has a wavelength of 113 nm with the emission of the analyte. 

The amount of AM was selected with respect to the visual detection and method sensitivity. Satisfactory visual detection was achieved in the fluorescent color change from blue to green corresponding to the Se concentration change in 5.0~125.0 μg/L. At the same time, full attention was paid to not causing the dilution of the solution by minimizing the volume of added AM solution.

The emission spectra of Se-DAN solutions and Se-DAN + AM solutions with varying quantities of Se are presented in Figure 2a and Figure 3a. Se-DAN exhibited a single emission peak at 523 nm, while Se-DAN + AM showed two peaks at 410 nm and 523 nm when excited at 376 nm light. The Se-DAN solution presented a clear green color under 365 nm UV illumination, as shown in Figure 2a inset. An increase in fluorescence intensity and a change in color, consistent with the spectral changes, were observed in the Se-DAN solutions, as the Se content increased from 5.0 to 125.0 μg/L. In the inset of Figure 3a, blue fluorescence from AM was visible in the Se-DAN + AM solution when illuminated with 365 nm UV light in the absence of Se or at very low concentrations. Increasing Se content from 5.0 to 125.0 μg/L resulted in a color change from blue to green and an increase in fluorescence at 523 nm while maintaining constant fluorescence at 410 nm. The fluorescence intensity of AM remained stable and served as a reference signal for ratiometric fluorescence, which forms the basis of Se visualization. Figure 2b displays the linear relations between the fluorescence intensity of the Se-DAN and the concentration of Se in 5.0~125.0 μg/L. The fitted linear data can be expressed as: ΔI = 1.122C + 13.82 
where C represents the concentration of Se, ΔI is the fluorescence intensity, and the fitting parameter (R^2^) is 0.9998. The theoretical limit of detection (LOD) and quantification (LOQ), calculated as (3.3 SD)/slope and (10 SD)/slope [21], were calculated to be 0.0048 and 0.013 μg/mL. 

Figure 3b depicts the linear correlation between fluorescence ratios (I_523_/I_410_) and concentration of Se, ranging from 5.0 to 125.0 μg/L. The linear regression analysis generated an equation for the fitted data, expressed as:I_523_/I_410_ = 0.00368C + 0.06281

The R^2^ was 0.9990, revealing an excellent correlation. The calculated LOD and LOQ were 0.0016 and 0.0049 μg/mL, respectively. There was a good linear relationship between the Se concentration and fluorescence intensity ratio in this range. The results show that fluorescence methods are susceptible to external factors when using the relative fluorescence intensity of a single emission wavelength for quantification, as described by the research [23]. Ratiometric fluorescence is not sensitive to environmental and background interferences, with self-correction capability, anti-interference, and improved detection sensitivity. Similarly, the wet digestion sample pretreatment for fluorometric determination has the issue of producing a high fluorescence blank value. However, the ratiometric fluorescence method is able to overcome this issue, provide more accurate quantification results, and avoid the false positive or false negative of single emission fluorescence. Furthermore, the intensity ratio of the two spectral bands is independent of the excitation intensity, thus allowing a reliable quantitative interpretation of the properties of the surroundings [28].

### 3.3. Comparison of Method Precision

Experiments were conducted under identical experimental conditions, and the data are presented in Table 1. The conventional fluorescence method exhibited inter-day and intra-day RSDs ranges of 4.05~5.26% and 2.59~3.09%, respectively, whereas the ratiometric fluorescence method demonstrated lower RSDs ranges of 2.08~2.78% and 1.28~1.84%. Results clearly indicate that the ratiometric fluorescence method surpasses the conventional fluorescence method in terms of both inter- and intra-day precision. Furthermore, our findings suggest superior intra-day precision compared with inter-day precision. Compared with the conventional fluorescence method, the ratiometric fluorescence approach offers excellent stability, instrumental reproducibility, and self-correction capabilities, as described by the research [29]. These features efficiently minimize the impact of various environmental factors on fluorescence determinations, thus differentiating it from conventional fluorescence methods.

### 3.4. Visual Detection Using Smartphone

In this work, a smartphone application (color recognizer) was used as a signal collector for the visual detection of Se(IV) (Figure 4). The fluorescent color information (RGB value) of the solutions was obtained by the smartphone application for Se(IV) analysis. As shown in Figure 5a,b, there was a linear relationship between the Se(IV) concentration and the G/B ratio. Finally, the concentration of Se(IV) was quantified by calculating the ratio of the green and blue color values (G/B). As shown in Figure 5a, G/B values showed a linear relationship with Se(IV) concentration at the first three points in the low concentration ranges. However, the linearity of the conventional fluorescence method in the G/B was poor, ranging from 5.0 to 125.0 μg/L (G/B = 0.2140C + 0.3851, R^2^ = 0.9108). The LOD and LOQ of the method were calculated to be 0.0045 and 0.015 μg/mL.

Figure 5b showed that G/B values of ratiometric fluorescence increase with the increase in Se(IV) concentration, which showed a good linear relationship with Se(IV) concentration, ranging from 5.0 to 125.0 μg/L (G/B = 0.0083C + 0.1823, R^2^ = 0.9960). The LOD and LOQ of the method were calculated to be 0.0040 and 0.012 μg/mL. The results showed that the ratiometric fluorescence method is favorable for the visual detection of Se.

### 3.5. Sample Analysis

The method developed has been applied to determine Se(IV) in selenium-enriched milk, resulting in average contents of 0.138 μg/mL by the conventional method and 0.181 μg/mL by the ratiometric method, as presented in Table 2. Both methods were also employed to determine the Se content of the shampoo samples, as listed in Appendix A. Se contents were determined to be 6.63 mg/mL and 8.83 mg/mL by the conventional and ratiometric methods, respectively. Sample pretreatment may introduce errors and affect the precision. There was minimal discrepancy in the RSD of the outcomes when comparing the two methods. This is because both the conventional fluorometric method and the ratiometric fluorometric method used the same digestion solution. Simultaneously, the digests were analyzed by ICP-MS to evaluate the method’s accuracy. It was found that the Se concentration in selenium-enriched milk measured via ICP-MS was 0.193 μg/mL. The results from the ratiometric fluorescence approach were close to those of ICP-MS, thus demonstrating that quantifying Se in selenium-enriched products using the ratiometric fluorescence method yields more accurate estimates of Se amount. Notably, compared with the ICP-MS results, the ratiometric fluorescence method’s ease of use and cheaper instrumentation make it a more viable option with a broader range of potential applications. Appendix A displays the reproducibility and precision results of the milk assay. In this study, we demonstrated that the regular fluorescence method and the ratiometric fluorescence method had RSDs of 11.9% and 5.13%, respectively. While the national standard method allows RSDs as high as 20%, our proposed method achieved an RSD of 5%, resulting in high reproducibility and accuracy. To further confirm these results, the selenium concentrations in the selenium-enriched milk samples were determined via microwave digestion [30], resulting in values of 0.189 and 0.193 μg/mL, respectively.

Selenium quantification of milk samples was performed by visual G/B analysis using the conventional fluorescence method, giving a result of 0.154 μg/mL with an RSD of 11% (n = 3). The visual G/B analysis of the ratiometric fluorescence method yielded a result of 0.176 μg/mL with an RSD of 5.9% (n = 3), which matched much better the ICP-MS results than the conventional fluorescence method, indicating that the ratiometric fluorescence method is promising means for visual determination of selenium.

### 3.6. Recovery of the Method

Furthermore, a sample of shampoo was employed for the spiked recovery experiment, and the results are shown in Table 3. The conventional fluorescence method achieved recoveries of 85.2~91.2% with an RSD of 3.6%. In contrast, the ratiometric fluorescence method achieved recoveries of 93.2~98.0% with an RSD of 2.5%. The experimental results demonstrate that the recoveries of the ratiometric fluorescence method are superior to those of the conventional fluorescence method, and the ratiometric fluorescence method was successfully applied to the determination of selenium content. A comparison was also made with previous literature (Table 4), and it was found that the LOD and LOQ were lower with the ratiometric fluorescence method.

### 3.7. Sample Precisions

The shampoo samples were measured every hour and one day apart to evaluate the RSDs, and the results are shown in Table 5. The conventional fluorescence method exhibited inter-day and intra-day RSD ranges of 4.66~8.09% and 4.60~12.5%, respectively, whereas the ratiometric fluorescence method demonstrated lower RSD ranges of 0.43~3.01% and 2.45~7.12%. The results clearly indicate that the ratiometric fluorescence method is superior to the conventional fluorescence method in terms of both inter- and intra-day precision. These features are due to the effective minimization of environmental interferences by the ratiometric method.

## 4. Conclusions

In this study, a ratiometric fluorescence assay using a single reference fluorescence reagent was developed for the determination of Se in selenium-enriched products. Ratiometric fluorescence offers higher precision and eliminates interference from environmental factors compared with conventional fluorescence, as evidenced by the lower RSD. The obtained results are consistent with those of ICP-MS. Ratiometric fluorescence-based visualization was successfully applied to the detection of selenium using a portable smartphone platform with excellent sensitivity and real-time detection capabilities. We propose a visual detection method for Se and will contribute to the development of a miniaturized detection device utilizing mobile phone image processing software in the future.

## Figures and Tables

**Figure 1 sensors-23-09187-f001:**
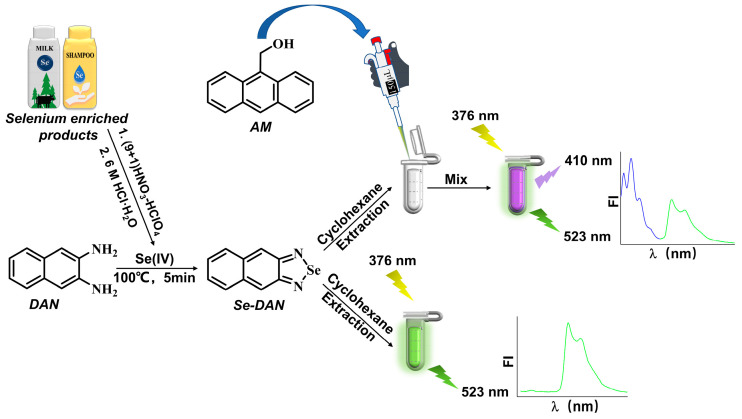
Scheme of the fluorometric (National Standard) and ratiometric fluorometric methods for the determination of Se(IV).

**Figure 2 sensors-23-09187-f002:**
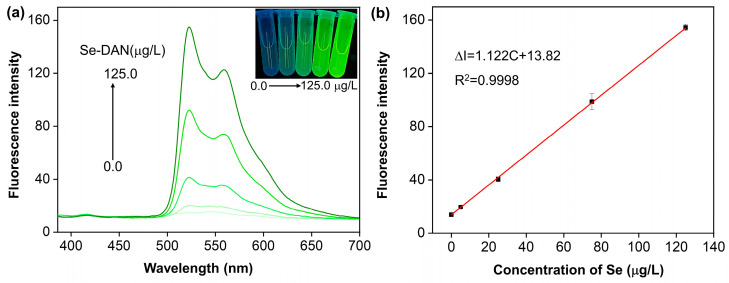
(**a**) Fluorescence emission spectra of the Se-DAN solution containing different concentrations of Se(IV) (0.0, 5.0, 25.0, 75.0, and 125.0 μg/L) in cyclohexane solution; inset picture shows the fluorescent color change in the Se-DAN with the different masses of Se(IV) under 365 nm light. (**b**) The relationship between fluorescence intensity and the concentrations of Se(IV).

**Figure 3 sensors-23-09187-f003:**
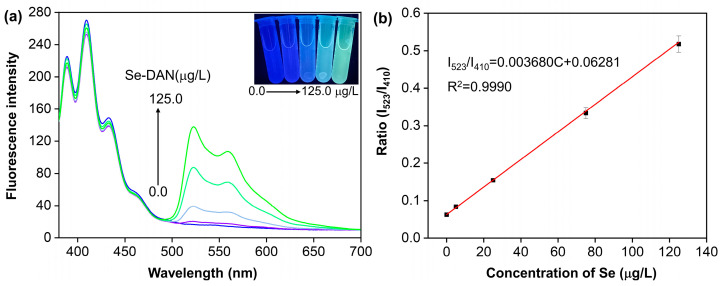
(**a**) Ratiometric fluorescence emission spectra of the Se-DAN + AM solution with the different concentrations of Se(IV) (0.0, 5.0, 25.0, 75.0, and 125.0 μg/L) in cyclohexane solution; the inset picture shows the fluorescent color change in the Se-DAN solutions with the different masses of Se(IV) under 365 nm light. (**b**) The relationship between the I_523_/I_410_ value and the concentrations of Se(IV).

**Figure 4 sensors-23-09187-f004:**
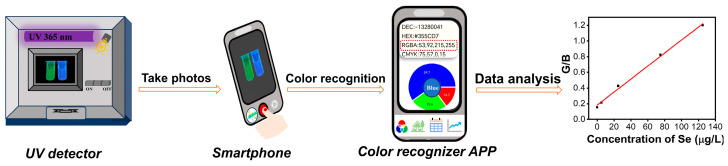
Schematic diagram of RGB analysis by the color recognizer of the smartphone.

**Figure 5 sensors-23-09187-f005:**
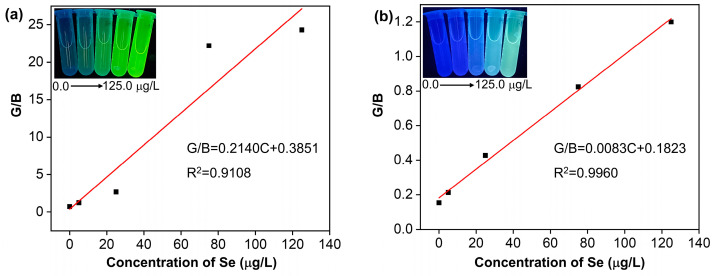
(**a**) Linear relationship plots of G/B values of conventional fluorescence versus Se(IV) concentrations; (**b**) Linear relationship plots of G/B values of ratiometric fluorescence versus Se(IV) concentrations. Insets: the fluorescence photographs were taken by a smartphone in the sensing platform under a 365 nm UV lamp.

**Table 1 sensors-23-09187-t001:** Inter-day and intra-day precision of the two fluorometric methods (n = 6).

Se (μg/L)	FL ^1^ (%)		rFL ^2^ (%)	
	Inter-Day	Intra-Day	Inter-Day	Intra-Day
25.0	5.26	2.97	2.32	1.28
75.0	4.05	2.59	2.08	1.54
125.0	4.18	3.09	2.78	1.84

^1^—fluorescence methods; ^2^—ratiometric fluorescence methods.

**Table 2 sensors-23-09187-t002:** Se measurement of result (μg/mL) in selenium-enriched milk.

Samples (n = 4)	FL	rFL	ICP-MS
1	0.139	0.177	0.173
2	0.134	0.183	0.182
3	0.134	0.175	0.209
4	0.144	0.189	0.209
Average	0.138 ± 0.005	0.181 ± 0.006	0.193 ± 0.019

**Table 3 sensors-23-09187-t003:** Spiked recovery experiments for the determination of selenium.

Original Content (μg/L)	Spiked (μg/L)	Detected (μg/L)	Recovery (%)	RSD (%)
26.0	5.0	30.5 ^1^	90.0	3.6
28.0	25.0	49.3 ^1^	85.2	
30.2	50.0	75.8 ^1^	91.2	
30.2	5.0	35.0 ^2^	96.0	2.5
28.0	25.0	52.5 ^2^	98.0	
30.0	50.0	76.6 ^2^	93.2	

^1^—fluorescence method; ^2^—ratiometric fluorescence method.

**Table 4 sensors-23-09187-t004:** Comparison of LOD and LOQ for the determination of Se(IV).

Sample	Method	Linear Range	LOD	LOQ	R^2^	Recovery	RSD	Ref.
		μg/mL				%		
Aqueous samples	UV–vis	0.20 ~3.00	0.150	0.520	0.990	99.0	6.94	[1]
Pharmaceuticals	MLC ^3^	0.330~3.30	0.102	0.309	0.9991	100	<1.00	[4]
Biological samples	UV–vis	15.0~35.0	1.32	3.99	0.999	98.8~99.5	<0.68	[16]
Water, vegetable	Colorimetry	1.0~6.00	0.050	-	0.993	94.1~108	<7.60	[17]
Cell culture solution	FL	1.0~10.0	0.190	-	0.9913	-	-	[19]
Milk, Shampoo	FL	0.005~1.25	0.0048	0.013	0.9998	85.2~91.2	3.6	[24]
Milk, shampoo	rFL	0.005~1.25	0.0016	0.0049	0.9990	93.2~98.0	2.5	This work

-—Not detected; ^3^—Micellar liquid chromatography.

**Table 5 sensors-23-09187-t005:** Inter-day and intra-day precisions of shampoo sample determination by the two methods.

Samples (n = 3)	FL ^1^ (%)		rFL ^2^ (%)	
	Inter-Day	Intra-Day	Inter-Day	Intra-Day
1	6.48	4.60	0.43	3.39
2	4.66	9.31	1.89	2.45
3	8.09	12.5	3.01	7.12

^1^—fluorescence methods; ^2^—ratiometric fluorescence methods.

## Data Availability

All data generated or analyzed during this study are included in this published article (and its Appendix A).

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
