# Peer review of "Determination of Selenium in Selenium—Enriched Products by Specific Ratiometric Fluorescence"

_sensors, 2023, doi:10.3390/s23229187_

Round 1

Reviewer 1 Report

Comments and Suggestions for Authors

The author wishes to report a ratio-metric fluorescence assay to the determination of selenium in selenium-enriched products. To establish the strategy, the authors have synthesized 2,3-diamino naphthalene (DAN) as a fluorescent ligand probe. The ratiometric method was compared with the conventional method in respect of precision and accuracy. This method was successfully applied to the determination of total selenium in selenium-enriched milk and selenium-supplemented shampoo with the results in agreement with those obtained by ICP-MS.

Overall, the work and the reported data are promising for subsequent work in improving the efficiency of fluorescence assay to the determination of selenium in selenium-enriched products. Thus, this work is acceptable for publication in this journal. A minor revision should be performed to address a few specific comments.

Specific Comments:

1.      This manuscript should be thoroughly revised in terms of grammatical mistakes and punctuation.

2.      References given in the manuscript body should match with the references list properly (for example 24 and 25).

3.      The author should check the determination of selenium in biological sample, water, and vegetable sample.

4.      The authors have proposed a visual detection method for Se which will contribute to the development of a miniaturized detection device utilizing mobile phone image processing software in the future. Author may attempt such experiments in this contribution also which will improve the quality of this manuscript.

Comments on the Quality of English Language

 This manuscript should be thoroughly revised in terms of grammatical mistakes and punctuation.

Author Response

  1. This manuscript should be thoroughly revised in terms of grammatical mistakes and punctuation.

This has been modified in the manuscript.

  1. References given in the manuscript body should match with the references list properly (for example 24 and 25).

All references have been checked and verified for relevance to the content of the manuscript.

  1. The author should check the determination of selenium in biological sample, water, and vegetable sample.

In our previous work, we have measured selenium in a variety of foods such as honey and oats, and did not detect selenium due to the low or no selenium content in foods in our region. Recently, we have gathered a range of samples from different regions and are presently conducting tests in the laboratory to detect selenium levels in selenium-enriched soil and tap water samples, and this method will be applied to more actual samples.

  1. The authors have proposed a visual detection method for Se which will contribute to the development of a miniaturized detection device utilizing mobile phone image processing software in the future. Author may attempt such experiments in this contribution also which will improve the quality of this manuscript.

This has been done in the article with additional experiments and analysed data as required.

Reviewer 2 Report

Comments and Suggestions for Authors

Overview and general recommendation: 

This manuscript developed a radiometric fluorescence assay for determining Se in selenium-enriched products with high precision and low limitation of determination. Generally, the manuscript is well-written, and experiments are well-planned. However, some questions need to be addressed carefully before the manuscript can be accepted. Some comments have been shown below.

Major Comments:

1. The background should be presented in the Abstract section.

2. Why is 365 nm chosen as the excitation wavelength? How to decide the excitation wavelength should be shown.

3. Compared with the inserted images in Figure 2a and Figure 3a, it seems to not easily observe the color change from blue to green.

4. As described ‘The results show that fluorescence methods are susceptible to external factors…’ in the Standard curve section, these LOD and LOQ results cannot elucidate the susceptibility of fluorescence to external factors distinctly.    

5. What is the mechanism for the detection of Se?

6. It is necessary to exhibit some evidence to confirm the successful synthesis of Se-DAN.

Author Response

  1. The background should be presented in the Abstract section.

      This has been modified as required in the manuscript.

  1. Why is 365 nm chosen as the excitation wavelength? How to decide the excitation wavelength should be shown.

     The excitation wavelength of the liquid phase ratiometric fluorescence method mentioned in the article is 376 nm, and the maximum emission wavelength of the Se-DAN molecule is 523 nm, and the maximum excitation wavelength of the excitation spectrum scanned at this wavelength is 382 nm. The reason for choosing 376 nm as the excitation wavelength was because the excitation wavelength of the second method of the National Standard of the People's Republic of China is 376 nm, and the second reason was to excite the ratiometric reagent at the same time because the emission spectrum of the ratiometric reagent is more complete under the excitation of 376 nm. Therefore, 376 nm was chosen as the excitation wavelength. The wavelength of the excitation source of the UV detector is 365 nm, and the wavelength of the excitation source of many portable UV lamps on the market is also 365 nm. 365 nm UV lamps were used as the excitation source to achieve visual detection of selenium and miniaturization of the instrument.

  1. Compared with the inserted images in Figure 2a and Figure 3a, it seems to not easily observe the color change from blue to green.

        The inset in Figure 2a shows only the green fluorescence of Se-DAN, so there is no blue fluorescence. The fluorescence diagram in Fig. 3a is the fluorescence diagram of the ratiometric fluorescence method, which has the blue fluorescence of the ratiometric fluorescent reagent and the green fluorescence of Se-DAN. The solution shows only blue fluorescence when there is no selenium in the solution, and changes from blue to green when the selenium content increases. This facilitates the visual detection of selenium.

  1. As described ‘The results show that fluorescence methods are susceptible to external factors…’ in the Standard curve section, these LOD and LOQ results cannot elucidate the susceptibility of fluorescence to external factors distinctly. 

           The conventional fluorescence method with a single emission wavelength is not only susceptible to interference by environmental factors, but also affected by residues such as nitric acid in the digestion solution, which leads to a high fluorescence blank value, and thus a high detection limit. Ratiometric fluorescence method has dual-emission fluorescence signals, and when disturbed by environmental factors, the two fluorescence signals are high or low at the same time, but the ratio of the two fluorescence signals remains unchanged, thus weakening the interference caused by environmental factors, and thus the detection limit of the ratiometric fluorescence method is lower than that of the conventional fluorescence method.

  1. What is the mechanism for the detection of Se?

        The 2,3-diaminonaphthalene (DAN) molecule is non-fluorescent, and it reacts with Se(IV) to form a fluorescent molecule of 4,5-phenylbenzoselenoxazole (Se-DAN), and the fluorescence intensity of Se-DAN is correlated with Se(IV) in the case of excess DAN participation in the reaction and shows a linear relationship. With reference to the above reaction characteristics and linearity, selenium can be quantitatively detected.

  1. It is necessary to exhibit some evidence to confirm the successful synthesis of Se-DAN.

Successful synthesis of Se-DAN was confirmed by NMR measurement of the synthetic product after purification by recrystallization in petroleum ether, and more detailed information can be found in the revised manuscript.

Round 2

Reviewer 2 Report

Comments and Suggestions for Authors

This manuscript has been improved based on comments provided by reviewers. It can be meet the requirements of publication in the Sensors.